# A Review of Recent Advances in Cognitive-Motor Dual-Tasking for Parkinson’s Disease Rehabilitation

**DOI:** 10.3390/s24196353

**Published:** 2024-09-30

**Authors:** Xiaohui Tan, Kai Wang, Wei Sun, Xinjin Li, Wenjie Wang, Feng Tian

**Affiliations:** 1Institute of Artificial Intelligence Education, Capital Normal University, Beijing 100048, China; 2Information Engineering College, Capital Normal University, Beijing 100048, China; 2231002087@cnu.edu.cn; 3Institute of Software, Chinese Academy of Sciences, Beijing 100045, China; sunwei2017@iscas.ac.cn (W.S.); lixinjin2022@iscas.ac.cn (X.L.); ashleywwj@163.com (W.W.); tianfeng@iscas.ac.cn (F.T.)

**Keywords:** Parkinson’s disease, rehabilitation, cognitive-motor dual-task, computer-assisted, virtual reality, exergames

## Abstract

Background: Parkinson’s disease is primarily characterized by the degeneration of motor neurons, leading to significant impairments in movement. Initially, physical therapy was predominantly employed to address these motor issues through targeted rehabilitation exercises. However, recent research has indicated that cognitive training can enhance the quality of life for patients with Parkinson’s. Consequently, some researchers have posited that the simultaneous engagement in computer-assisted motor and cognitive dual-task (CADT) may yield superior therapeutic outcomes. Methods: A comprehensive literature search was performed across various databases, and studies were selected following PRISMA guidelines, focusing on CADT rehabilitation interventions. Results: Dual-task training enhances Parkinson’s disease (PD) rehabilitation by automating movements and minimizing secondary task interference. The inclusion of a sensor system provides real-time feedback to help patients make immediate adjustments during training. Furthermore, CADT promotes more vigorous participation and commitment to training exercises, especially those that are repetitive and can lead to patient boredom and demotivation. Virtual reality-tailored tasks, closely mirroring everyday challenges, facilitate more efficient patient adaptation post-rehabilitation. Conclusions: Although the current studies are limited by small sample sizes and low levels, CADT rehabilitation presents as a significant, effective, and potential strategy for PD.

## 1. Introduction

Parkinson’s disease (PD) is classified as a progressively deteriorating neurological condition prominently defined by its quintessential motoric manifestations, which encompass slowed movement (bradykinesia), tremors at rest, increased muscle stiffness, and impaired balance (postural instability). In addition to these motor manifestations, PD is also associated with nonmotor symptoms, including cognitive impairments, which further complicate the lives of those affected. The ability of individuals with PD to perform activities of daily living is often compromised, with these challenges intensifying as the disease advances. Although pharmacological interventions for PD can provide some relief from motor symptoms, the management of gait and balance issues remains a significant challenge for patients throughout the disease’s trajectory [1].

In everyday life, the capacity to multitask is crucial, as it often necessitates the concurrent execution of cognitive and motor tasks. This skill is fundamental to a wide array of daily activities, enabling individuals to engage in complex behaviors such as navigating their environment while remaining vigilant to potential hazards and obstacles during ambulation. The interplay between cognitive processes and motor actions is integral to performing efficiently and safely in a variety of settings [2,3]. The performance of these tasks relies on the harmonized operation of motor and cognitive systems. Rehabilitation offers a feasible path for bolstering the physical and cognitive capabilities of individuals with Parkinson’s disease (PD) [4]. Motor-cognitive dual-task training, a strategy that transcends disciplinary boundaries, merges exercises that target motor skills with those that enhance cognitive abilities. This approach is designed to simultaneously cultivate and hone both the body’s physical capacities and the mind’s cognitive functions, therefore maximizing the efficiency of functional performance and the cohesion of cognitive-motor coordination [5]. Dual-tasking inherently involves the concurrent execution of two distinct goal-oriented activities, such as solving arithmetic problems concurrently with walking. Therefore, in individuals with PD, performance on motor-cognitive tasks is typically diminished relative to healthy peers matched for age, sex, and education [6]. Concurrently, the simultaneous activation of both motor and cognitive functions within the dual-task framework can yield further advantages in comparison to single-task training protocols [7].

Although Cognitive-Motor Dual-Tasking (CMDT) can be implemented with or without the support of technological systems [8], the integration of sophisticated technological interfaces in CMDT promotes the simultaneous conduct of motor and cognitive tasks. These interfaces are instrumental in monitoring participant performance, delivering real-time feedback, and enhancing patient engagement [9,10]. An increasing corpus of pilot and feasibility studies has demonstrated the efficacy of motion-tracking systems within CMDT training for PD. These systems adeptly mimic the patient’s movements within the context of immersive virtual reality (VR) scenarios [11,12,13,14,15] or on digital screens [16,17,18,19], concurrent with the engagement in cognitive exercises as part of the interaction with such technologies [9,20,21,22]. Specifically, fully immersive VR setups, including head-mounted displays and automated virtual environments akin to caves, provide the means to generate authentic cognitive challenges. These tasks simulate real-world interactions on the body and provide multisensory experiences within the virtual environment [23,24]. Figure 1 illustrates that the integration of sensors and computer technology in dual-tasking represents the current approach to computer-assisted motor-cognitive dual-task training (CADT).

A significant body of review has explored the effects of dual-task training (DTT) on improving gait, balance, motor symptoms, and cognitive functions in individuals with PD [2,8,25,26,27,28,29]. Nonetheless, to our best understanding, no systematic review has yet to narrowly focus on examining the specific contributions of CADT for PD rehabilitation within this demographic.

The research methodology is comprehensively detailed in Section 2, which encompasses an extensive literature review, the selection of appropriate databases, the identification of relevant search terms, and the criteria for study inclusion. Following this, Section 3 delves into the application of DTT in the neurorehabilitation of PD. This chapter highlights how DTT enables PD patients to navigate the complexities of daily life more effectively by enhancing both motor automation and cognitive processing. Subsequently, we introduce methods for constructing CADT, which include traditional DTT, the integration of sensors within the intervention, and the significance of computer-assisted training. Our discussion culminates in an evaluation of the effectiveness and feasibility of CADT for PD rehabilitation, particularly emerging technologies in recent years.

## 2. Methods

This chapter provides a detailed overview of the research methodology, covering the literature search process, database selection, identification of search terms, and criteria for study selection.

### 2.1. Literature Search

An extensive literature search was conducted using a strategic selection of databases, including Web of Science, PubMed, Elsevier, and MDPI. This selection aimed to ensure a comprehensive acquisition of relevant data from a wide range of medical and engineering journals pertinent to our research interests. The search strategy employed specific terms such as “Parkinson’s disease”, “dual-task training”, “cognitive dysfunction”, “walking functions”, and “computer-assisted”. Two independent researchers meticulously executed this process to maintain objectivity and rigor in the study selection.

The thoroughness of the search extended not only to the databases utilized but also to the sample sizes of the included studies, which provided robust evidence for our analysis. In instances of disagreement between the initial evaluators, a third impartial examiner was engaged to reach a consensus, therefore enhancing the reliability of our literature review process. Following a careful categorization and assessment of the obtained studies, we consolidated the literature. This consolidation aimed to clarify the effects of dual-task training on cognitive deficits and motor capabilities in individuals with Parkinson’s disease within the framework of PD rehabilitation. Ultimately, this process sought to provide a nuanced understanding of the impact of these interventions based on extensive data gathered from a diverse pool of subjects.

### 2.2. Selection Criteria

We conducted our systematic search by the PRISMA guidelines [30]. To maximize the breadth of our search, the initial query was designed to be as inclusive as possible, utilizing the keyword ’dual task’ within the title, abstract, and keywords (TAK) of the literature. The exclusion criteria were as follows: (1) publications not in English, (2) those published before 2013, (3) research not involving human participants, (4) non-original full-text articles, and (6) works not in harmony with the aims of the present investigation (i.e., Some studies have a propensity to concentrate on the verification of algorithms and the categorization processes involved in machine learning, frequently overlooking the exploration of gait and balance features and metrics, as well as research into alternative wearable technologies).

The progression of treatment modalities for Parkinson’s disease has evolved from single-task training, including cognitive and motor training, to dual-task training, as illustrated in Figure 2. In recent years, there has been a gradual increase in research into the use of computer-assisted devices (VR, exergames) during interventions to improve the effectiveness of the treatment or the accuracy of diagnostic assessments. These may be included in future studies. The use of brain-computer interfaces and exoskeleton-assisted devices in telemedicine and personalized medicine.

## 3. Advantages and Enhancement of Dual-Task Training

This chapter examines the application of dual-task training (DTT) in the neurorehabilitation of Parkinson’s disease (PD), highlighting its role in enhancing motor automation and cognitive processing. By improving these aspects, DTT enables PD patients to better navigate the challenges of daily life.

### 3.1. Mechanism of Dual-Task Training

Dual-task training (DTT) is regarded as a central strategy for the neurorehabilitation of Parkinson’s disease (PD), a disorder that progressively affects the nervous system and is marked by the intricate relationship between cognitive and motor dysfunctions [1,38]. In PD, cognitive functions are closely intertwined with motor capabilities. A key mechanism underlying DTT’s physical benefits is the enhancement of motor automaticity, which emphasizes the importance of interventions that target both domains [39]. The ability to execute skilled motor tasks with minimal conscious attention or executive control [40]. The theoretical foundation of DTT’s effectiveness is grounded in the allocation and scheduling hypothesis, proposing that DTT enhances the distribution of cognitive resources and the organization of task execution during multifaceted activities [25,41]. This leads to not only enhanced motor automaticity but also improved cognitive processing, equipping individuals with PD to better navigate daily life challenges.

The majority of DTT protocols described in scholarly works entail the concurrent performance of one physical task alongside one verbal cognitive task [26]. This approach recognizes PD’s multifaceted nature and the need for a comprehensive rehabilitation strategy [42]. Furthermore, DTT’s success is contingent upon the continuous adaptation of training protocols to meet participants’ changing needs, ensuring ongoing relevance and effectiveness.

In summary, DTT is a promising neurorehabilitation avenue for individuals with PD, offering a dual-pronged approach targeting motor automaticity and cognitive enhancement [43]. Through the amalgamation of these components, DTT holds the promise of markedly enhancing the quality of life for individuals with PD, offering a comprehensive and flexible strategy to tackle the multifaceted challenges presented by the disease [44]. Figure 3 illustrates the theoretical foundation for employing physical methods in the treatment of PD.

### 3.2. Dual-Task Practice Advantage

In addition to the improvements attributed solely to motor training, an additional benefit arises from combining motor and cognitive training, known as the dual-task practice advantage [45,46,47]. This advantage denotes the enhanced ability to perform in dual-task conditions after engaging in the dual-task practice, rather than single-task practice where each task is carried out in a stepwise fashion [48]. The existence of a dual-task practice advantage has been supported by evidence from a randomized controlled trial that included a population of older adults [49].

Single-task training (STT) traditionally involves focusing on one activity at a time, often used in rehabilitation to improve specific motor skills [50]. Unlike dual-task training, which challenges individuals with Parkinson’s disease (PD) to perform both cognitive and motor tasks simultaneously, STT does not impose the same cognitive load [26]. While STT can lead to improvements in the targeted motor function, it may not address the cognitive-motor interference often experienced in daily activities by those with PD [27]. Previous studies have shown that DTT can promote the automation of movement and reduce the interference of secondary tasks with the execution of primary motor tasks in individuals with Parkinson’s disease (PD), unlike STT [28].

Most of the studies have focused on assessing improvements in cognitive and motor domains following intervention. Key areas of interest include the reduction of falls, enhancement of balance, optimization of gait parameters, amelioration of dual-task performance, and the sharpening of executive functions and attention.

#### 3.2.1. Balance and Fall

Falls are prevalent and frequently lead to severe incapacitation among individuals with Parkinson’s disease (PD), resulting in considerable adverse outcomes [51]. The frequency of falls is notably higher in the PD population, with individuals experiencing falls at a rate double that of their non-PD counterparts [52]. The ramifications of falls are multifaceted, encompassing bone fractures, hospitalization, and a decline in mobility, all of which can severely affect the quality of life [53,54]. The repercussions of these events, when compounded by the fear of falling, can initiate a vicious cycle that increases the likelihood of further falls [55]. The economic impact of fall-related injuries is also considerable, imposing substantial costs on both patients and healthcare systems [56]. Given the gravity of these repercussions, the incidence of falls has transformed into a significant public health concern. Consequently, it is of paramount importance to concentrate efforts on the development and implementation of strategies aimed at the prevention and management of falls within the Parkinson’s disease (PD) population [29,57]. This endeavor necessitates the precise identification of risk factors inherently associated with PD that contribute to the propensity for falls.

Furthermore, deficits in dual-tasking have been identified as a significant risk factor for falls in individuals with PD, as demonstrated by cross-sectional studies [58]. This highlights the significance of incorporating dual-task training as a critical component within rehabilitation programs for PD, as well as in the broader spectrum of fall prevention strategies. In a study, The metric of the area under the curve (AUC) was employed to assess the Falls Efficacy Scale-International (FES-I). Several gait parameters, assessed under both single-task and dual-task walking scenarios, indicated substantial disparities in performance [59]. Notably, the foot strike angle stood out with the highest AUC values in both conditions: 0.728 for single-task walking (with a cut-off at 14.07°) and 0.742 for dual-task walking (with a cut-off at 12.53°). In the context of single-task walking, other parameters exhibiting high AUC scores encompassed the variability in trunk transverse range of motion, stride length, lumbar transverse range of motion, single limb support variability, and the duration of turns. When it came to dual-task walking, the gait metrics that also registered significant AUC values were cadence variability, single limb support variability, stride length, double support variability, and the number of steps in a turn.

#### 3.2.2. Gait Function

For individuals with Parkinson’s disease (PD), the restoration of walking ability is a paramount concern [60]. Among the various facets of walking behavior that engage clinicians, gait speed, stride length, and cadence have been identified as critical target symptoms. The significance of gait speed, stride length, and cadence in the clinical assessment of Parkinson’s disease (PD) cannot be overstated. These metrics are crucial for gauging community independence, predicting key health outcomes [61,62], and understanding their correlation with mortality rates [63]. Often considered vital clinical indicators, these gait characteristics are closely tied to the severity of PD as evaluated by various clinical scales. Gait hypokinesia in PD shows a substantial correlation with scores from the Unified Parkinson’s Disease Rating Scale (UPDRS), including total scores and those related to activities of daily living, as well as scores from the Columbia University Rating Scale for bradykinesia, and the UPDRS-III axial motor scores. Additionally, they reflect limitations in physical activity and an overall increase in disability. The importance and practicality of using gait speed, stride length, and cadence as clinical indicators hinge significantly on their sensitivity to change—meaning, their ability to accurately mirror changes that occur over time [64]. Recognizing the responsiveness of these gait aspects to treatment is crucial for evaluating the effectiveness of interventions designed to address gait impairments in PD. While gait parameters of individuals with PD can be compared as a percentage relative to normative values adjusted for age and sex, there remains a scarcity of benchmarks that indicate what constitutes a clinically significant change in gait performance for those with PD [65]. Nonetheless, these gait characteristics are consistently utilized as primary outcomes in rehabilitation efforts and studies concerning pharmacological and surgical treatments for gait disorders. Additionally, they are recognized as reliable predictors of an individual’s ability to walk independently within the community.

Walking Speed: Participants in the dual-task (DT) group showed a marked increase in gait velocity after the statistically significant intervention, and this enhancement was maintained throughout the follow-up period under various conditions. Conversely, the single-task (ST) group only displayed significant improvement under the motor condition, with the benefits lasting through the follow-up. Across all post-rehabilitation and follow-up assessments, the DT group consistently demonstrated higher gait velocity compared to the ST group, with these differences being statistically significant.Stride Length: Post-rehabilitation, there was an increase in stride length for both groups under the same conditions and time frames that were used to evaluate gait velocity, resulting in similar differences between the groups. When comparing the conditions, the dual-task (DT) group consistently showed a longer stride length in the single-task (ST) condition as opposed to the verbal and auditory dual-task conditions in all assessments.Cadence: After the rehabilitation period, the dual-task (DT) group recorded a notable uptick in step count in the single-task (ST) scenario, an advancement that was still evident at the time of follow-up. Nevertheless, no substantial improvements were tracked in the dual-task settings, except for the auditory task, where an uptick in cadence was detected exclusively during the follow-up phase, reflecting the ST group’s performance in the ST condition. Significant disparities in cadence between the two groups were observed solely in the ST condition post-rehabilitation and the auditory and motor conditions during the subsequent assessment period.

#### 3.2.3. Cognitive Function

Cognitive function, particularly executive function, and attention plays a pivotal role in mobility, making rehabilitation interventions that target both cognitive enhancement and motor improvement potentially very beneficial for mobility outcomes [17,40,66,67,68]. Incorporating a cognitive element into physical training sessions may reduce the utilization of compensatory cognitive strategies among participants, instead promoting greater dependence on the striatal motor pathway to execute movements. While the effects of dual-task training on cognitive functioning in individuals with Parkinson’s disease (PD) warrant further investigation [69,70], recent research has indicated the potential benefits of integrated approaches that tackle both physical and cognitive aspects concurrently [71]. A randomized controlled trial (RCT) documented that individuals who underwent dual-task training showed significant improvements in cognitive performance during dual-task walking, a marked contrast to those in the group that received no intervention [72]. In another single-blind RCT involving 40 patients with Parkinson’s disease (PD), the dual-task training group showed a non-significant trend towards better performance in executive function tests, such as the Frontal Assessment Battery and Trail-Making Test, after training when compared to the single-task training group that concentrated solely on gait training [73].

The dual-task group exhibited significant enhancements in both gait velocity and stride length across all post-training assessment conditions, alongside improvements in perceived quality of life. In contrast, the single-task group observed improvements in these gait parameters solely under the motor condition following the intervention, without any significant advancement in perceived quality of life. Additionally, comparative analyses revealed that the dual-task group maintained higher velocities and stride lengths compared to the single-task group post-treatment, irrespective of the assessment conditions.

The discussion encompasses the underlying mechanisms and practical advantages of DTT and its potential effects on balance, gait function, and cognitive abilities in individuals with PD. Furthermore, the chapter emphasizes the benefits of DTT in reducing fall risk and enhancing these patients’ overall quality of life. However, it also addresses several limitations that warrant further investigation, including small sample sizes, insufficient long-term follow-up data, and challenges related to the applicability of DTT to specific patient populations.

## 4. Construction of Motor-Cognitive Dual-Task Training

This chapter outlines the methodologies for developing motor-cognitive dual-task training (CADT), emphasizing traditional dual-task training practices, the integration of sensors in interventions, and the significance of computer-assisted training. It highlights the role of wearable sensors and virtual reality in enhancing the rehabilitation experience for individuals with Parkinson’s disease (PD), therefore improving the personalization and motivational aspects of training. Furthermore, the chapter explores the potential benefits of motor games and virtual environments in enhancing patient engagement, executive function, and balance. Finally, it suggests directions for future research, focusing on technology acceptance, implementation contexts, and the assessment of long-term outcomes.

The core methodology for creating motor-cognitive dual-task training entails overlaying a cognitive challenge onto a physical movement task [74]. This cognitive element can vary in complexity, starting from simple activities like arithmetic problem-solving to more complex ones, such as integrating verbal fluency tasks with recitation and memory exercises. The motor tasks can fluctuate in complexity, from basic gait training to more demanding programs like HiBalance [75]. The training’s intensity is crafted to be adaptable, increasing progressively as participants become accustomed to initial difficulty levels and go on to exceed their prior achievements. Additionally, the use of electronic devices within the training program brings a host of advantages to the regimen. The fusion of motor and cognitive tasks can be facilitated through exergames that utilize video or virtual reality, therefore injecting an element of entertainment and convenience into the training process [40]. Transportable exercise gear, such as stationary cycles and monitoring sensors, broadens the scope of rehabilitation, allowing it to transcend the limits of medical facilities and be conducted within the convenience of one’s home.

The following Figure 4 illustrates the progression of dual-tasking interventions in Parkinson’s treatment. The first phase represents traditional treatment methods, as discussed in Section 4.1: Traditional Dual-Task Training. In the second phase, sensors were integrated into the intervention, detailed in Section 4.2: Sensors in Intervention. The third phase involves the increasing application of computer-assisted interventions, which is explored in Section 4.3: Computer-Assisted Dual-Task Training.

### 4.1. Traditional Dual-Task Training

Traditional dual-task training typically integrates straightforward physical exercises with cognitive challenges to enhance the ability to perform multiple tasks simultaneously [76,77,78,79,80,81,82]. For example, a patient might be asked to walk while reciting the alphabet in reverse or to tap their fingers to a steady beat while counting numbers aloud [49]. This method targets the improvement of motor functions and cognitive skills, such as attention and executive functions, which are often impacted in individuals with Parkinson’s disease. By practicing these tasks together, the training aims to foster better coordination between cognitive and motor systems, thus potentially improving overall daily functioning and reducing fall risk [44,83,84,85,86].

Current academic pursuits are focused on exploring the therapeutic potential of various interventions aimed at individuals with Parkinson’s disease (PD), prioritizing the enhancement of walking ability, cognitive functions, and general health. A key study conducted by reported on a randomized controlled trial that demonstrated the greater effectiveness of dual-task group training over single-task group training for this patient group [87]. This dual-task approach, which combines simultaneous cognitive and motor exercises, led to significant improvements in walking speed, stride length, and the participants’ perceived quality of life. In contrast, the single-task training group experienced benefits only in motor aspects, without notable changes in cognitive functioning or perceived quality of life.

Building upon these findings, a pilot study to assess the specific effects of cognitive and motor dual-task gait training on the execution of dual-task gait in individuals with PD [88]. The study found that cognitive dual-task gait training (CDTT) significantly decreased the time spent in double support during cognitive dual tasks and resulted in improved motor dual-task walking performance, demonstrated by increased gait speed, longer stride length, and reduced double support time. On the other hand, motor dual-task gait training (MDTT) was particularly effective at decreasing variability in gait during motor dual tasks. This research highlights the unique but complementary advantages of both CDTT and MDTT for the rehabilitation of individuals with PD.

In addition to the aforementioned studies, therapeutic climbing (TC) has gained recognition as an integrative method within neurological rehabilitation, providing a comprehensive workout that may positively influence both the physical and psychosocial faculties of people with Parkinson’s (PwP) [89,90,91,92,93]. Nonetheless, the current empirical support for the targeted advantages of TC in PwP is constrained, underscoring the necessity for additional research to discern how such activities could be seamlessly woven into PD rehabilitation protocols to foster a well-rounded enhancement in both motor and cognitive spheres [70,94,95,96].

### 4.2. Sensors in Intervention

Recently, wearable inertial sensors have gained popularity for assessing mobility due to their compact size, lightweight design, portability, and affordability [97,98,99]. Coupled with user-friendly software, these sensors can be seamlessly integrated into clinical practice, providing immediate outcomes upon the assessment’s conclusion. This ease of use and swift feedback make wearable inertial sensors a valuable tool for both clinical and research applications in mobility assessment [100,101,102,103,104]. As wearable technologies continue to advance, as illustrated in Figure 5, it is essential to conduct thorough clinical validation against established scales for assessing Parkinson’s disease (PD) symptoms.

Over the past few years, wearable technologies have risen as valuable instruments for delineating the symptoms of Parkinson’s disease (PD), providing a deeper and more detailed insight into the progression of these symptoms beyond what conventional rating scales can offerThese portable and non-invasive devices provide real-time monitoring of a variety of comorbidities, allowing for more detailed insights into the motor and nonmotor pathology of PD. The analysis of spatiotemporal features captured by wearables is crucial for enhancing clinical trial assessments and for optimizing treatment regimens in a time-efficient manner, as noted by a study [13].

Wearable sensors, particularly inertial measurement units (IMUs), have proven invaluable for monitoring the longitudinal progression of motor symptoms [109,110,111,112]. These devices can record continuous kinematics in various settings, both within and outside clinical environments. IMUs, equipped with accelerometers, gyroscopes, and magnetometers, are typically attached to different parts of the body, such as the feet, shanks, and knees, to capture the nuances of PD movement. A study utilized IMUs in conjunction with convolutional neural networks to track freezing of gait (FOG) in PD patients, identifying optimal sensor locations to maximize patient adherence for at-home monitoring. Furthermore, smartphones equipped with three-dimensional accelerometers have been employed to measure spatiotemporal features of rest tremors.

Expanding beyond motor symptoms, electroencephalography (EEG) allows for the quantitative assessment of cognitive states through event-related potentials (ERPs) [113,114,115,116]. These event-related potentials (ERPs), calculated as the mean electroencephalogram (EEG) activity synchronized with sensory triggers, shed light on cognitive operations like assessing stimuli and preparing responses. Cognitive impairment in Parkinson’s disease (PD) is marked by specific ERP traits, like an increased P300 amplitude at 300 ms post-stimulus, showing sensory processing difficulties. Additionally, higher alpha power in power spectral density points to issues in memory load management, as documented by [117,118,119,120,121,122].

Clinically validated wearables will enhance PD diagnosis and improve patients’ quality of life [123,124,125,126,127]. With the rapid development of medical technology, advanced sensing technologies such as flexible electronics [128,129], wearable microfluidics [130], and hyperstretch strain sensors [131,132] have opened up new possibilities for treating Parkinson’s disease (PD). It is through this ongoing process of innovation and validation that the potential of wearables in PD management can be fully realized, offering hope for more effective treatments and a better understanding of this complex condition [128,132,133,134,135]. The choice of BCI platform is crucial and depends on various factors, including research goals, equipment cost, and patient comfort levels. Figure 6 illustrates several different types of EEG measurements. MEG (Magnetoencephalography) is a high-precision platform that provides excellent spatial and temporal resolution, albeit with high costs and the need for specialized facilities. EEG (Electroencephalography) is known for its high temporal resolution and non-invasive nature, EEG is the most commonly used platform in BCI research and applications. With the highest resolutions, ECoG (Electrocorticography) is an invasive technique used primarily in clinical settings for its direct cortical contact. fNIRS (Functional Near-Infrared Spectroscopy) is a non-invasive technique that offers a portable and relatively cost-effective alternative for brain imaging with good spatial resolution.

### 4.3. Computers-Assisted Dual-Task Training

Recent scientific advancements have shed light on the potential of interactive and technology-based interventions to augment motor and cognitive functions in individuals with Parkinson’s disease (PD). The graph in Figure 7 illustrates various scenarios in which PD engages in computer-assisted cognitive-motor dual tasks.

The Table 1 below presents a selection of studies relevant to this section. These studies were chosen because they all involve dual-task training utilizing computer-assisted technology. Additionally, they specifically target Parkinson’s disease (PD). The primary method for measuring outcomes in these studies is the analysis of patients’ gait information. This selection allows for comparative analyses, facilitating a better understanding of the advantages and disadvantages of existing research in this area.

#### 4.3.1. Exergames

In recent times, there has been a surge in interest in game-based interventions owing to their capacity to harness motivation and engagement, therefore enhancing rehabilitation results for clinical groups [140,141,142]. The appeal of gamification is primarily due to the rewarding experiences it offers through multisensory engagement, interactive elements, and entertaining game design, all of which are key aspects of the rehabilitation journey [19,143,144]. Consequently, the therapeutic advantages of rehabilitation can be increased by promoting more vigorous participation and commitment to training exercises, especially those that are repetitive and can lead to patient boredom and demotivation [139]. Individuals who are more motivated and engaged are more inclined to follow through with training regimens and remain in the program, which in turn can decrease the rates of program discontinuation [145,146,147,148,149].

Van Beek et al. [16] investigated the merits of exergaming-based dexterity training in PD. They utilized games of graduated difficulty that targeted activities such as number imitation, block connection, pedal removal from a flower, rope cutting, and dot connection, all designed to augment dexterity and cognitive faculties. In a pioneering effort, Szturm et al. [17] suggested a multi-center, randomized controlled trial to investigate how a treadmill system featuring dual-task cognitive games affects the metabolic activity and gait function in people with Parkinson’s disease (PD). This trial is significant for its plan to use behavioral positron emission tomography (PET) for brain imaging, which could provide a deeper understanding of the molecular and neural mechanisms behind the decline in mobility and cognitive functions, as well as to assess the effectiveness of the intervention. Mishra et al. [18] introduced an innovative instrumented trail-making task (TMT) designed to evaluate motor and cognitive performance in PD. This iTMT, which necessitates participants to sequentially touch indexed circles while maintaining an upright position, has demonstrated high feasibility and has proven adept at distinguishing between cognitively intact older adults, PD patients, and those with mild cognitive impairment (MCI) based on task completion times. Broadening the research horizon.

Continuing the exploration of exergaming’s benefits, Jäggi and colleagues [139] conducted a randomized controlled trial to explore the viability and effectiveness of cognitive-motor exergaming as a supplementary treatment for Parkinson’s disease in patients during rehabilitation. The study yielded high adherence rates, an absence of adverse events, and notable improvements in both cognitive and motor domains, therefore underscoring the exergaming’s potential effectiveness in this demographic.

Continuing in this vein, Bhatt et al. [19] scrutinized the test-retest reliability and validity of a computerized dual-task (DT) assessment using a rehabilitation treadmill platform (GRP) in PD patients. The GRP exhibited moderate to high intraclass correlation coefficient (ICC) values for spatiotemporal gait measures and visuomotor and cognitive game performance measures, indicating its reliability in assessing DT interference effects and monitoring disease progression in PD.

Lastly, De Melo Cerqueira et al. employed a unique approach, carrying out a quasi-experimental controlled study to evaluate the effects of Kinect-based gaming on the cognitive and motor capabilities of individuals with Parkinson’s disease (PD), as opposed to a group of healthy senior citizens. The results pointed towards enhancements in executive functions following motor-cognitive training with Kinect games. However, the sustainability of these gains was not pronounced in the PD group, and no significant motor improvements were detected.

The studies referenced above are illustrated in Figure 8. Within the context of Parkinson’s disease (PD), game technologies originally designed for recreational use, such as Microsoft Kinect and Nintendo Wii, have significantly improved both motor and cognitive performance [139,150]. The benefits derived from these technologies are deemed to be as advantageous as those obtained through standard rehabilitation techniques [88]. In addition, certain game technologies have been tailored for training to home in on specific motor and cognitive areas [17,151]. Nonetheless, the use of such technologies as rehabilitation aids for the delivery of dual-task training (DTT) in PD neurorehabilitation remains largely uncharted or is restricted in its capacity to technically adapt to the varied and changing training requirements of individuals in real time [16,17,152,153]. At present, there is inconsistent evidence concerning the superior performance of different exergame platforms [154].

#### 4.3.2. Virtualization Environment (VE)

The incorporation of immersive technologies such as augmented reality (AR) and virtual reality (VR) is an emerging area with substantial prospects for the research and rehabilitation of gait and balance difficulties in individuals who have Parkinson’s disease (PD) [121,156,157]. These advanced technologies provide a platform for users to immerse themselves in complex, enriched environments that can be intricately personalized to align with each user’s unique needs and capabilities [158]. Within the realm of PD assessment, AR and VR have proven particularly useful in manipulating virtual environments to facilitate a more profound exploration into the behavioral and neural foundations underpinning gait and balance [12,111,159,160]. This methodological innovation has been instrumental in advancing our comprehension of the intricate motor-cognitive neural circuitry associated with PD [161].

In recent years, the integration of immersive technologies into the rehabilitation of Parkinson’s disease (PD) has emerged as a promising therapeutic approach. A body of research has investigated the potential of such technologies to enhance motor and cognitive functions, which are often impaired in individuals with PD.

Delving deeper into immersive technology, Yun et al. [11] investigated the practicality of employing fully immersive virtual reality (VR) exergames that incorporate dual-task elements for the treatment of patients with Parkinson’s disease (PD). The study found that these VR exergames were well-tolerated by the participants and showed promise in improving executive function and balance without causing adverse effects. This indicates that VR exergames could be a viable rehabilitation tool for PD.

Expanding on this concept, Alberts et al. [13] detailed the creation of the Dual-task Augmented Reality Treatment (DART) platform. This platform leverages the Microsoft HoloLens2 AR device to deliver dual-task training to PD patients. The DART platform was successful in inducing dual-task interference, a common challenge for PD patients, and received high usability ratings from users, suggesting its effectiveness in addressing gait and postural instability.

In a related approach, Pullia et al. [12] assessed the impact of treadmill training combined with semi-immersive VR on PD patients. Participants in the experimental group, who underwent training with the C-Mill system, demonstrated notable enhancements in several motor assessment metrics relative to the control group. These findings indicate that VR-augmented treadmill training may present a potent therapeutic approach for enhancing gait and balance capabilities in individuals with Parkinson’s disease (PD). Lau et al. [14] have delved into the feasibility and preliminary efficacy of an immersive treadmill training system that incorporates a first-person video game. This intervention was designed to target gait and cognition in patients with early to mid-stage PD. The results were encouraging, with the intervention group demonstrating significant improvements in gait speed, walking distance, and cognitive performance. These findings underscore the potential of immersive technology to play a pivotal role in PD rehabilitation.

Lastly, Pelosin et al. [15] have investigated the effects of motor-cognitive treadmill training with virtual reality (TT + VR) on gait performance, cognitive functions, and fall risk in PD patients. The research indicated that an extended 12-week TT + VR training regimen resulted in greater advancements in cognitive functions and a more pronounced decrease in the incidence of falls compared to a 6-week program.

In summary, the studies illustrated in Figure 9 highlight the therapeutic potential of immersive technologies in the rehabilitation of Parkinson’s disease (PD). These studies demonstrate that virtual reality (VR) and augmented reality (AR) interventions can significantly improve both motor and cognitive functions. Additionally, they enhance patient engagement and may reduce the risk of falls [93]. As these technologies evolve, they promise personalized and efficient rehab solutions for those with PD. While VR is believed to be more effective and resource-efficient than non-VR strategies, firm evidence is still limited. Nonetheless, VR’s capacity for customized programs tailored to motor learning principles offers a notable benefit [162]. These customizable interventions hold promise for deployment across both clinical and domestic spheres, with the innate flexibility to evolve in tandem with the shifting requirements of individuals over time. The successful design and execution of these technological systems demand a synergistic collaboration between a diverse array of stakeholders, encompassing researchers, clinicians, technology developers, and patients [163,164,165]. Adopting an interdisciplinary approach is paramount to ensuring the optimal usability, engagement, safety, and overall efficacy of immersive AR and VR technologies as they are applied within the framework of PD rehabilitation [15].

## 5. Discussion

Recent advancements in computer-assisted dual-tasking methods for Parkinson’s disease (PD) demonstrate not only their viability and safety but also their efficacy. These innovative approaches have shown a greater degree of effectiveness compared to traditional rehabilitation and standard exercise protocols.

Exergames offer significant advantages over traditional rehabilitation methods, particularly in facilitating home-based rehabilitation and enabling remote monitoring. These features highlight the potential for exergames to play a crucial role in the evolving landscape of Parkinson’s disease (PD) rehabilitation. They provide patients with more accessible and engaging therapeutic options. Research supports the feasibility and emerging efficacy of virtual reality technology in gait and cognitive training for individuals with PD. Studies have shown that immersive virtual reality training can improve gait velocity and walking distances. In some cases, it may also enhance cognitive functions. Additionally, the integration of virtual reality with dual-task training has demonstrated the potential to improve executive functions and balance in patients.

While these studies have achieved notable outcomes, they are not without limitations:Initially, the modest participant numbers, absence of extended follow-up information, and restricted applicability to particular groups could limit the universal relevance and statistical robustness of the conclusions. For instance, in studies concerning immersive virtual reality gait training, only a small number of participants involved participants primarily from specialized outpatient clinics, which may not be representative of the general elderly population.Second, some studies lacked a control group, or the design of the control group may not have been rigorous enough, affecting the comparability and persuasiveness of the results. Additionally, some studies failed to control for confounding variables; they did not account for all potential confounders, such as medication adjustments, disease severity, and “on/off” states, which could influence the interpretation of the intervention effects.Furthermore, the intensity and duration of the training in some studies may not have been sufficient to produce significant clinical effects. Longer training periods and more intensive interventions are needed to confirm the durability and efficacy of the effects.There are also considerations regarding the acceptance of technology, with some studies pointing out the need for more information on the acceptance of new technology-based training programs by both participants and clinicians. Additionally, the implementation environment of some studies limits their application in home or community settings. For example, the strict protocols used in clinical environments may be challenging to replicate in home settings.

In summary, while the studies have provided valuable insights, there is a need for large-scale, long-term studies with rigorous control group designs that account for confounding variables and consider the practical aspects of technology acceptance and real-world application in home and community settings.

Figure 10 shows our vision regarding future interventions that the future of dual-mission is very promising. It is imperative that subsequent research endeavors not only furnish more stringent clinical trials to affirm the hypotheses introduced but also concentrate on modifying current therapeutic strategies for domestic use and refining data acquisition methods. This may involve the use of a finger sensor to track hand tremors [166]. Other metrics, including dysphonia [167] and variations in heart rate [168], might also prove to be pertinent for investigation. Assessing the viability of broadening the scope of PD Exergame therapy to encompass patients with mild cognitive impairment and Parkinson’s-related dementia, along with the integration of pioneering interaction modalities such as detailed motor exercises and vocal command systems, are crucial areas for further exploration. In conclusion, therapeutic initiatives should be individualized to suit the particular physical and cognitive traits of each patient.

Future research direction:Future research should build upon the existing foundation by validating the effectiveness of technology-assisted training through larger-scale clinical trials. Additionally, it is essential to explore its applicability across diverse patient populations. Individualized treatment is crucial for improving outcomes in Parkinson’s disease, as each patient’s symptoms, disease progression, and treatment responses are unique. By collecting and analyzing a patient’s physical, psychological, and behavioral data, healthcare professionals can develop more accurate and effective personalized rehabilitation plans. Furthermore, personalized treatment enhances patient engagement in the therapeutic process, leading to improved compliance and satisfaction.Also, researchers need to develop new assessment tools to more accurately measure training effects, such as functional magnetic resonance imaging (fMRI), which can be used in conjunction with neuroimaging to observe the effects of brain-computer interaction therapy on the patient’s brain activity and to explore its effectiveness from the perspective of neural mechanisms.Then, exploring how these training methods can be better integrated into patients’ daily lives to achieve sustained rehabilitation outcomes. With the development of smart healthcare technologies, it is expected that more innovative treatments will emerge in the future, such as brain-computer interfaces, panoramic views, metaverse, and telemedicine, which will be more precise and personalized and be able to provide customized treatment plans based on the patient’s specific situation.Future research should prioritize the integration of patients’ physical, psychological, and social needs to deliver comprehensive support. Additionally, enhancing awareness and trust in new technologies among patients and healthcare professionals is crucial. Research should also explore ways to optimize technology usage, ensuring compatibility with patients’ actual needs and lifestyle habits. Furthermore, effective integration of these technologies into home and community settings is necessary to improve accessibility and utility. Cost-effectiveness and patient affordability must be key considerations in this process.

## 6. Conclusions

Computer-assisted cognitive-motor dual-task training (CADT) represents a cutting-edge rehabilitation strategy for Parkinson’s patients. It leverages real-time feedback to tailor the difficulty of exercises to individual performance, ensuring an optimal challenge level. Personalized programs cater to each patient’s unique needs, while the incorporation of wearable sensors and virtual reality enriches the rehabilitation experience, boosting motivation and engagement.

Our study reveals significant benefits of applying computer technology to dual cognitive-motor tasks for Parkinson’s disease rehabilitation. In addition, our findings investigate the integration of the latest CADT( e.g., VR, exergames) with patients’ daily lives and its potential impact in family and community settings. While current studies have provided valuable insights, there is a need for large-scale, long-term studies that incorporate rigorous control group designs. These studies should also account for confounding variables and consider the practical aspects of technology acceptance and real-world application in home and community settings.

As the field of smart healthcare advances, the potential for CADT to revolutionize Parkinson’s disease treatment is substantial, promising a surge in research and the development of more effective therapeutic options.

## Figures and Tables

**Figure 1 sensors-24-06353-f001:**
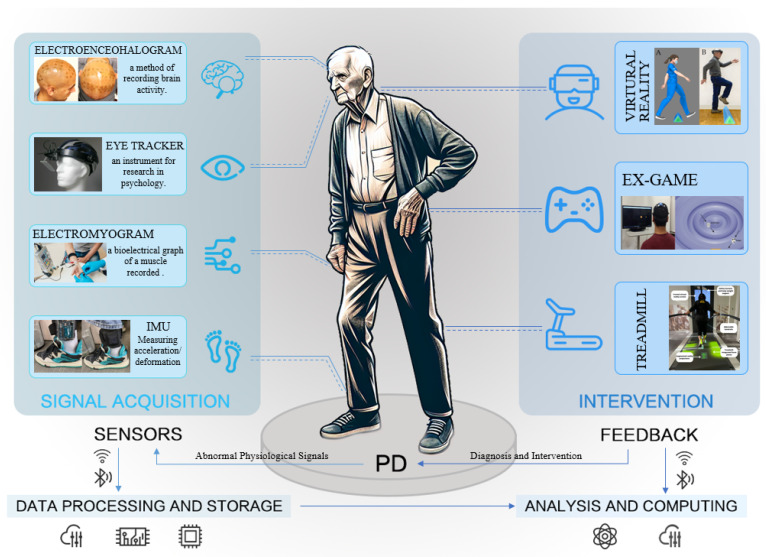
Wearable and intervention systems for PD, classification, and related technologies.

**Figure 2 sensors-24-06353-f002:**
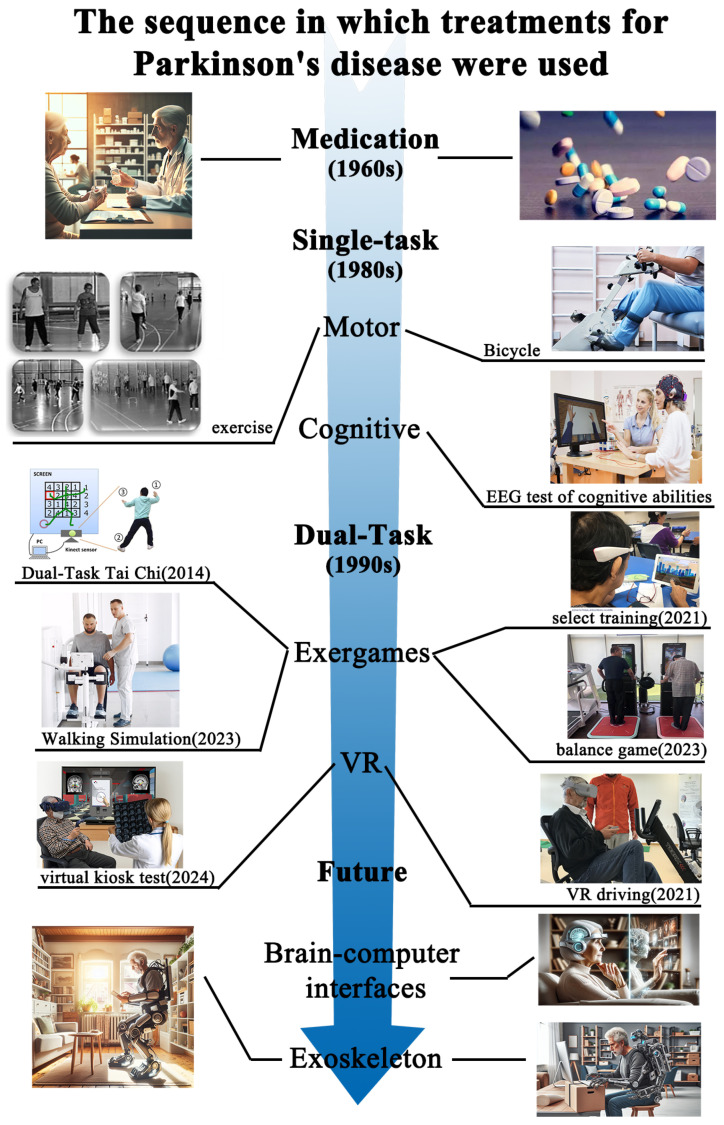
The sequence in which treatments for Parkinson’s disease were used [31,32,33,34,35,36,37].

**Figure 3 sensors-24-06353-f003:**
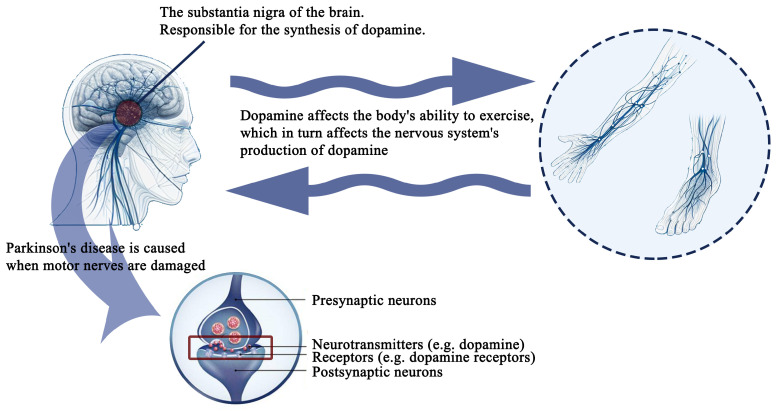
The relationship between the brain and the parts of the body.

**Figure 4 sensors-24-06353-f004:**
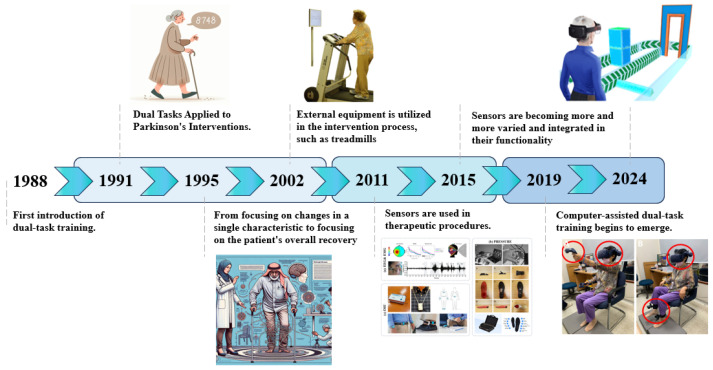
The development of dual-task training in the treatment of Parkinson’s interventions. The first phase is the traditional treatment phase. In the second phase, sensors were utilized in the intervention, and in the third phase so far, computer-assisted interventions have been progressively applied [13].

**Figure 5 sensors-24-06353-f005:**
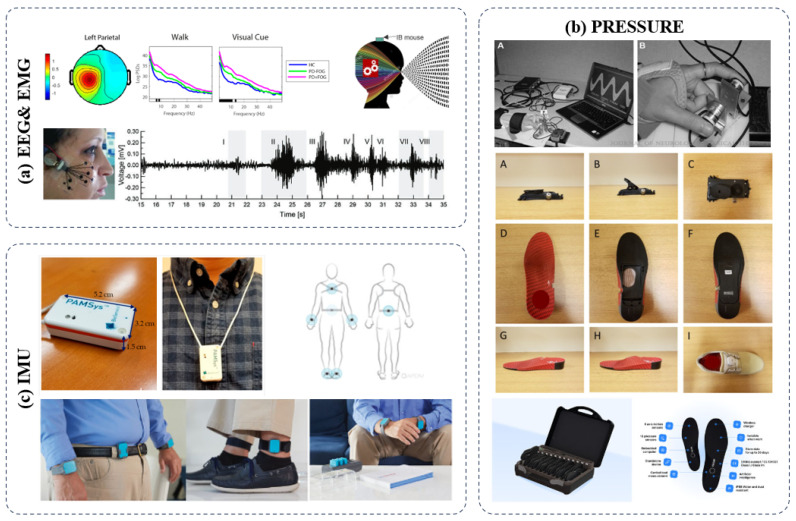
Types of sensors used during Parkinson’s interventions. EEG tests cognitive ability, EMG and pressure sensors, and IMU assesses motor ability [105,106,107,108].

**Figure 6 sensors-24-06353-f006:**
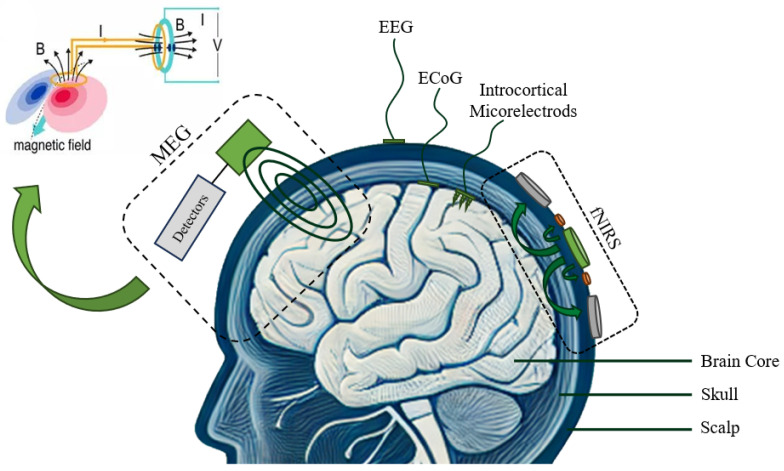
Types of BCI sensor mounting include invasive (IM), semi-invasive (ECoG), and non-invasive methods (MEG, EEG, fNIRS).

**Figure 7 sensors-24-06353-f007:**
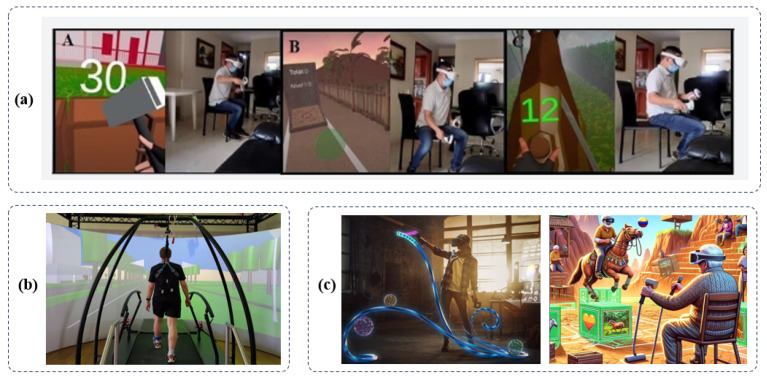
Interventions of selected papers. (**a**) Exergames using VR for gaming workouts (**b**) Treadmill-assisted training using semi-immersive environments (**c**) Cognitive and motor training using VR in immersive virtual environments [136,137,138].

**Figure 8 sensors-24-06353-f008:**
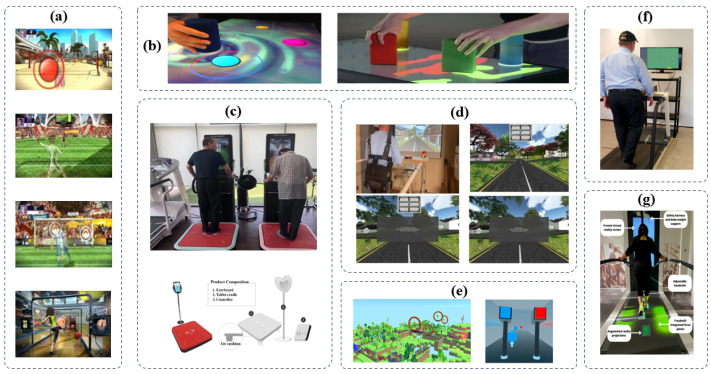
Exgames intervention processing. (**a**) Kickball games, (**b**) Tracking Ball Games. (**c**) Balance Control Games, (**d**) Treadmill Simulation Outdoor Games, (**e**) Hammer Tapping Game (**f**) Treadmill training with external screen (**g**) Treadmill training with indicator lights [36,155].

**Figure 9 sensors-24-06353-f009:**
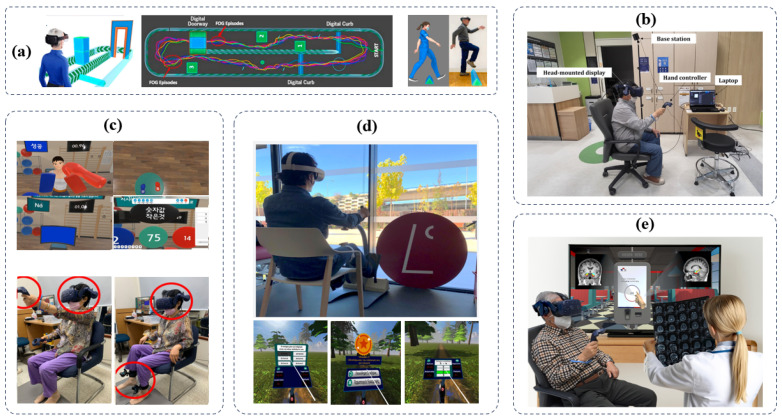
Virtual reality intervention processing. (**a**) DART platform. (**b**) VR shot. (**c**) VR boxing. (**d**) VR cycle. (**e**) VR cognitive test [32].

**Figure 10 sensors-24-06353-f010:**
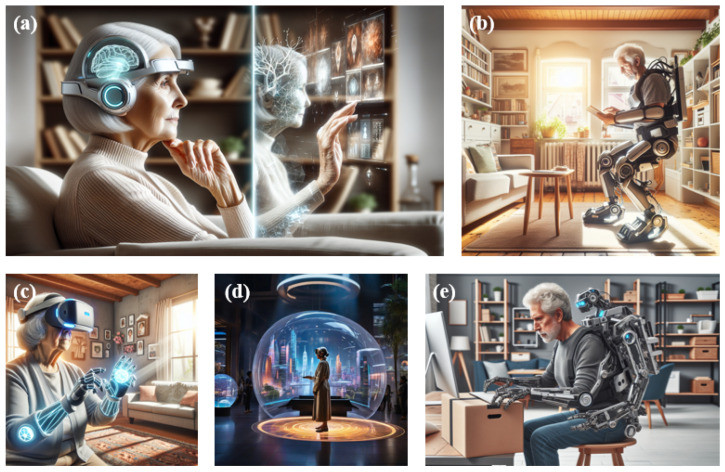
Possible future interventions. (**a**) brain-computer interface, (**b**,**e**) exoskeleton, (**c**,**d**) metaverse.

**Table 1 sensors-24-06353-t001:** These studies have employed dual-task training with computer-assisted techniques specifically for Parkinson’s disease (PD), measuring results primarily through the analysis of patients’ gait information.

Study	Sample	Sensors	CADTa Technology	Main Results	Limitations
Yun et al. [11]	8 PD	The HTC Vive Pro*	VR exergames	BBS tests improved; TUG time saw no significant changes; SSQ scores showed no significant differences.	Small sample, no control group, VR game intensity inadequate for outcomes, and larger RCTs for validation.
Alberts et al. [13]	48 PD	Microsoft HoloLens2 AR*	Dual-task Augmented Reality platform	Decreases in gait metrics were noted in both single and dual-task obstacle performances.	DART’s long-term effects and dual-task efficacy in PD need more study; cost and AR acceptance may limit adoption.
Pullia et al. [12]	20 PD	C-Mill: body weight sensors	The C-Mill system*	Post-treatment improvements in both groups confirmed physiotherapy’s benefits with EG’s VR-enhanced gait.	small sample, no efficacy analysis for size, no motion capture, requires larger RCT for validation.
Lau et al. [14]	22 PD	APDM Opal sensors*	immersive treadmill training	The treadmill program improved gait and walking in the experimental group without adverse effects, in contrast to the control group’s stable or declining condition.	Small sample, high dropout rate, pilot study limits, clinician-led progression, and missed participant feedback opportunities.
Pelosin et al. [15]	96 PD	Kinect^®^ NeuroTrax™*	TT + VR system (V-TIME)	Dual-task gait and cognition improved in both groups, with fewer falls and less fear after a 12-week program, indicating extended TT + VR training offers cognitive improvements.	Smaller 12-week sample, no active control, excludes dementia/severe cognitive impairment, unknown long-term training effects.
Van et al. [16]	8 PD (1 MCI)	IMU	SMARTfit^®^	Gamified DTT proved feasible, safe, and more effective than STT, though results varied by demographic and clinical factors.	Results limited to a small PD group with mild symptoms; impact on severe cases and general applicability unknown; further studies needed.
Szturm et al. [17]	52 PD	Physical Activity Tracking Image Acquisition	Game-Based Treadmill System	Gait-cognitive training may re-organize the prefrontal cortex, applicable to vascular impairment and Alzheimer’s, utilizing DT mobility training and PET imaging.	Study efficacy and applicability depend on broader PD validation, patient compliance, and retention.
Mishra et al. [18]	14 PD, and 11 MCI, 14 CN	LegSys*	instrumented trail-making task (TMT)	iTMT showed significant differences and effects among CN-older, PD, and MCI groups, correlating with gait speed and MoCA scores.	Sample not representative of elderly PD due to specialized clinic recruitment, older MCI group, different gait tools, and need for iTMT sensitivity research.
Jäggi et al. [139]	40 PD	strain gauges, 5 vibration motors.	Dividat Sensor	High adherence, minimal dropouts, no adverse events, and significant interaction effects on cognitive and physical measures underscore the intervention’s positive influence.	inpatient PD focus, brief intervention, COVID-19 reduced sample, bias risk, non-tailored exercise, limited measures, medication/disease confounders; future research needed.
Bhatt et al. [19]	30 PD	pressure mat IMU	a computer game-based rehabilitation treadmill platform (GRP)	The protocol had moderate to high reliability for gait and performance in ST and DT, with notable DT drops and differences between PD-3 and PD-2 groups.	Study constrained by no control group, treadmill vs. field walking, head rotation effects, unquantified attention, and small sample limiting generalizability.
De Melo et al. [18]	8 PD and 8 without PD	Microsoft Kinect™ sensor	Xbox 360 Microsoft Kinect™ commercial games	Xbox Kinect™ training enhances executive functions in PD patients and elders, but PD patients do not retain FAB test improvements.	Study limited by small sample, brief training, no control, vague criteria, and lack of detectable change and follow-up.

Taiwan, China. Washington, DC, USA. California, CA, USA. Philadelphia, PA, USA. San Diego, CA, USA.

## Data Availability

Data are contained within the article.

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
