# Peer review of "A Review of Recent Advances in Cognitive-Motor Dual-Tasking for Parkinson’s Disease Rehabilitation"

_sensors, 2024, doi:10.3390/s24196353_

Round 1

Reviewer 1 Report

Comments and Suggestions for Authors

Comments on Authors (General)

·       The presentation styles should be modified for publication.

·       The performance of all recent works should be mentioned in included studies in Table.1.

·       The repeated presentation must be omitted.

Comments on Authors (Specific)

·       The authors should present the recommendations on survey research and contributions of this study.

·       The brain-computer interaction is an outstanding approach for PD but the feasible studies shall be carried out for giving the recommendation in future works. More analyses must be done for confirmations.

·       More literature survey is needed for giving the recommendations and the limitations on technology and non-technology points of view must be presented.

·       The pros and cons of the feasible studies on traditional methods and innovative directions shall be presented.

·       The robustness of the recommended system must be discussed in detail.

Comments on the Quality of English Language

 Extensive editing of English language required

Reviewer 2 Report

Comments and Suggestions for Authors

For this review of computer-assisted motor and cognitive dual-task (CADT), the authors have found an impressive list of 208 references that constitute around 10 pages (of 27 pages of the article).

 The reviewer found this article difficult to follow.  Mainly because the authors do not explain how did they organized and combined those 208 references. I major work for the authors for such a review is to categorize the references in classes and comment them. That is probably what they have done, in the titles and subtitles, but what is missing is that they should explain usually in the introduction, explaining how is the work organized.  Usually each main section should be introduced in the introduction. After for each main section they should introduce how it is organized.

 Another major concern are Table 1 and 2 (same table). The legend says “Table 1. Descriptive table of the included studies.” With a footnote “The table includes only experimental studies and does not list the review articles mentioned in the text .” They incluse only refs. 11 to 21

 This is not clearly explained. It means that references 22-208 deal with non-experimental studies ???. What is the logic of citing all those references? This should be clearly explained and all the references grouped in tables or some other criteria.

 A review paper should organize references in tables such as Table 1,2 for the major part of references.

 It is also important that the authors discuss more the databases and number of subjects. There are incoherencies in their text: in Sect2.2 an exclusion criteria is “(5) case studies or studies with a participant number less than 10,” but in table 1 a study with 8PD subjects is cited.

Did the authors included review papers on the CADT task? It seems not and this could be interesting to see what is the positioning of their review paper concerning other reviews if they exist.

Comments on the Quality of English Language

Did authors used generative AI tools such as ChatGPT?

Regarding english:

Line 15: current studies instead of currently studies

 And typos ;Punctuations , such as Line 67 : (PD)[27? ,28].

 Line.15:  post-rehabilitation.Conclusions:

Line citations : « trajectory. [1] »

Line 195 ,  208, 234 …..

Reviewer 3 Report

Comments and Suggestions for Authors

This manuscript reviews the evalution of PD treatment with the focus on CMDT. The history of treatments, the contributions and promise of CMDT, the role of new technologies (e.g., sensors, VR, games, etc.) in PD treatment is covered extensively.  Some of my comments are as follows;

1) Even though they started contributing to the PD treatment fairly recently, IMU, Electrophysiological and pressure sensors have been around for decades. It would be helpful to briefly mention the future promise of more advanced and recent sensing technologies such as flexible electronics, wearable microfluidics, and ultrstretchable strain sensors for PD treatment. 

2) The reference formatting is consistently inconsistent throughout the manuscript. The brackets should be before the punctuation mark, I think. Also there should be a consistent space with the words and reference brackets.

3) The way references and figures are mentioned in the main text is inappropriate. For example the content of the references and figures are attributed to the numbers. Instead the content should be mentioned and the reference/figure number should be given separately. Example:

” In the study[73], Participants in the dual-task (DT) group showed…” should be “ In a study by X et al., participants … [73].

or

“ Looking ahead, with Figure 9 we can imagine that the future” should be “ Figure 9 shows our vision regarding future interventions…”

4) Finally, I am not sure about the mdpi policy on using AI-generated images and how they should be cited, but Figure 9 seems to be AI generated and I think there should be a certain way of handling this type of figures and how they are disclosed.

Comments on the Quality of English Language

Generally formatting and punctuation should be improved.

Round 2

Reviewer 1 Report

Comments and Suggestions for Authors

The paper is fine and accepted.

Reviewer 3 Report

Comments and Suggestions for Authors

The manuscript is revised satisfactorily, even though there are still some minor formatting issues persist.